# The impact of US Government Stop Work Order on HIV epidemic trajectory in Zimbabwe

**Isaac Taramusi**[1,2]*, **John Stover**[3], **Ali Feizzadeh**[1], **Ngwarai Sithole**[4], **Amon Mpofu**[5], **Tsitsi Apollo**[4], **Owen Mugurungi**[4], **Benard Madzima**[5], **London Makwanya**[5], **Henry Damisoni**[1], **Richard Makurumidze**[2], **Simbarashe Rusakaniko**[2]

1 Department of Programs, UNAIDS UCO Zimbabwe, Harare, Zimbabwe, 2 Department of Global, Public Health and Family Medicine, Faculty of Medicine and Health Sciences, University of Zimbabwe, Harare, Zimbabwe, 3 Modeling and Analysis Department, Avenir Health, Glastonbury, Connecticut, United States of America, 4 AIDS and TB Unit, Ministry of Health and Child Care, Harare, Zimbabwe, 5 Monitoring and Evaluation Department, National AIDS Council of Zimbabwe, Harare, Zimbabwe

* itaramusi@gmail.com

## Abstract

Zimbabwe is highly dependent on United States Government (USG) funding for the HIV Response, where 42% of the HIV expenditures in 2020 were financed by the USG and in 2023, PEPFAR invested more than $200 million to support HIV programmes. In late January 2025, the USG issued a "Stop Work Order," suspending funding for international aid and development during a 90-day review period and this was expected to have a significant negative impact on the Zimbabwean healthcare system therefore an epidemiological assessment was conducted to assess the impact of the stop work order. To assess the impact on incidence and mortality, an epidemiological assessment using the Avenir Health Goals model simulated five scenarios ranging from temporary pauses to the indefinite withdrawal of PEPFAR and 11% reduction in the Global Fund funding support. The stopping funding specified in the waiver plus 11% reduction in the Global Fund funding results in 74% (11,000) additional new HIV infections. The 90-day stop-work order will result in 35% (6,000) additional new HIV infections in 2025. The stopping of PEPFAR direct funding will result in additional 7,000 new HIV infections in 2025. Stopping all PEPFAR direct funding will add 35,000, 90-days pause will add 31,000, while stopping support to HIV prevention programs will add 22,000 additional new HIV infections through 2030. Stopping all PEPFAR direct funding will result in more than 13% increase in AIDS-related deaths with additional 107,000 deaths through 2030. The 90 days pause will reduce adult treatment coverage by 10% from 95% to 85% in 2025, while stopping direct PEPFAR funding will result in a reduction in adult ART coverage by 40% to 55% in 2025. These disruptions pose a significant threat to Zimbabwe's ability to achieve epidemic control and reach the 2030 target of ending AIDS as a public health threat.

**Data availability statement:** This is a modelling study and hence no new primary data were collected for this study. Data inputs for the model were publicly available online: Demographic data: https://zimbabwe.opendata-forafrica.org/anjlptc/2022-population-hous-ing-census, https://population.un.org/wpp/. Surveys: https://dhsprogram.com/publications/publication-fr186-dhs-final-reports.cfm, https://phia.icap.columbia.edu/zimbabwe2020-final-re-port/. Surveillance: https://www.mohcc.gov.zw/. Program data: https://www.mohcc.gov.zw/, https://aidsreportingtool.unaids.org/, https://nac.org.zw/. Goals model used is an open-source application found on website: https://www.avenirhealth.org/software-spectrum.html.

**Funding:** The author(s) received no specific funding for this work.

**Competing interests:** The authors have declared that no competing interests exist.

## Background

Zimbabwe is highly dependent on United States Government (USG) funding for the HIV Response, where 42% of the HIV expenditures in 2020 were financed by the USG [1] and in 2023, PEPFAR invested more than $200 million to support HIV programmes. The United States Government is a key contributor to the Zimbabwean healthcare system through PEPFAR (The U.S. President's Emergency Plan for AIDS Relief) and Global Fund to Fight AIDS, Tuberculosis and Malaria (Global Fund). PEPFAR is implemented through the United States Agency for International Development (USAID) and Center for Disease Control (CDC). PEPFAR has been the largest donor followed by the Global Fund from 2021-2024 contributing up to $850 million in healthcare resources [2]. The country has been implementing high-quality HIV prevention and treatment programmes supported by the U.S. Government over the past 20 years, including during the first Trump administration, that have been highly effective. The country has achieved some remarkable progress over the years. A key achievement has been reaching the 95-95-95 treatment targets (95% of people living with HIV know their status, 95% of them are on ART and 95% of them are virally suppressed) by reaching 95-98-96 in 2024. Zimbabwe also achieved 80% reduction in new HIV infections from 79,000 in 2010–15,000 in 2023 and 66% reduction in AIDS-related deaths from 57,000 in 2010–19,000 in 2023 [3]. The Vertical trans-mission rate also reduced from 23.79% in 2010 to 7.35% in 2023 [3]. The country managed to scale up treatment from 28% in 2010 to 95% in 2023 [3].

In late January 2025, the USG issued a "Stop Work Order," suspending funding for international aid and development during a 90-day review period. This effectively stopped key HIV activities funded under The U.S. Presidents Emergency Plan for AIDS Relief (PEPFAR). The stop order on USAID funding is expected to have a sig-nificant negative impact on the Zimbabwean healthcare system. Zimbabwe has lost at least 27% of total annual external funding for health in 2025 [2].

There are 1.3 million people living with HIV and PEPFAR provided HIV services for more than 860,000 in 2024. A total of 15,000 new HIV infections occurs annually so any Stop Work Order have the potential to severely destabilize the HIV response in the country. Although a waiver was issued on 10 February, allowing the continuation of key care and treatment services for the general public and prevention services for pregnant and breastfeeding women, PEPFAR-supported implementing partners have faced difficulties resuming service delivery due to reduced financial and human resources, specifically during the first three weeks the stop order was in effect before the waiver was issued. The PEPFAR supported partners were unable to plan for the future after the 90-days and therefore have had to make pre-emptive cuts to their programs and operations. The waiver largely left behind vulnerable and key popu-lations who have heightened risk for HIV transmission and low retention and poor treatment outcomes. The government of Zimbabwe put measures in place to ensure continued service provision.

An epidemiological assessment was conducted to assess the impact of the Stop Work Order on the HIV Response in the country. The modelling analysis was

therefore crucial to support planning and decision-making efforts by providing information to answer key sustainability questions.

## Methods

Using a validated published GOALS HIV impact model developed by Avenir Health we estimated the impact of the stop work order. The Goals application to Zimbabwe was validated by comparing the model estimates of prevalence over time by population group with available prevalence data from national surveys in 2005–06 (Demographic and Health Survey-DHS), 2010–11 (DHS), 2015–16 (DHS), 2015–16 (Population-based HIV Impact Assessment - PHIA) and 2020 (PHIA). Goals contains a transmission model that calculates the number of new HIV infections over time as a result of sexual and injecting drug transmission. It links to the AIDS Impact Model (AIM) model to also calculate new child infections due to mother-to-child transmission [4]. The Goals model framework and structure are outlined (see S1 and S2 Figs). Goals was used to investigate the impact of the stop work order by creating five scenarios; the base scenario where business will continue as usual, 90-day pause where programmes are expected to resume after 90 days, stopping PEPFAR funded HIV prevention programmes to time indefinite, stopping direct PEPFAR funding including treatment, and a scenario of stopping funding specified in the waiver plus 11% reduction in the Global Fund funding. All the scenarios assumed no mitigation from government and other donors to close the funding gap. The scenarios were agreed with National AIDS Council and Ministry of Health and Child Care AIDS and TB unit. The goals base file was updated with 2024 program data.

Secondary aggregate data was used as data input to the model. This includes demographic estimates from the United Nations World Population Projections since 1970 and census reports from 1982, 1992, 2002, 2012, and 2022. Routine program data since 2000 for HIV prevention, treatment, and care programmes was sourced from DHIS 2. These data include HIV prevalence from routine pregnant women testing at sentinel sites; the number or percent of adults and children receiving antiretroviral therapy (ART) and/or Cotrimoxazole since 2004; utilization of Prevention of Vertical transmission (PMTCT) programmes since 2002; percentage lost to follow-up for ART clients each year; ART retention at delivery; number of first ANC visits; number of pregnant women receiving at least one HIV test; number of pregnant women testing positive from the first HIV test; number of pregnant women known to be HIV positive at first ANC visit (including those on ART); number of pregnant women on ART prior to first ANC; ANC pregnant women re-tested for HIV during ANC; ANC and PMTCT trends over time; knowledge of status; viral suppression thresholds; and monthly number on ART by age and sex. Survey data will be drawn from ZDHS 2005, 2010, and 2015 for prevalence and HIV risk behaviour, and from ZIMPHIA 2015/16 and 2020 for incidence, prevalence, and HIV risk behaviour. Surveillance data from ANC rounds in 1989, 1991, 1993, 1995, 1997, 2000, 2002, 2004, 2006, 2009, 2012, and 2016 was also included.

Outcomes of interest were the number of HIV infections added through 2030, reduction in coverage of HIV treatment services and expected increases in mortality.

## Results

The stopping funding specified in the waiver plus 11% reduction in the Global Fund funding results 74% increase in new HIV infections, that is additional 11,000 infections in 2026 (Fig 1). The 90-day stop working order will result in 35% increase in new HIV infections, that is 6,000 additional infections in 2025. The stopping of PEPFAR direct funding will result in 45% increase in new HIV infections, that translate to additional 7,000 infections in 2025. Stopping HIV prevention programmes supported by PEPFAR will result in 19% increase in new HIV infections, that is 3,000 additional infections in 2025. Besides the status quo, all other scenarios will result in additional new HIV infections through 2030, Stopping all PEPFAR direct funding will add 35,000, 90-days pause will add 31,000, while stopping support to HIV prevention programs will add 22,000 additional infections through 2030 (Fig 2).

 

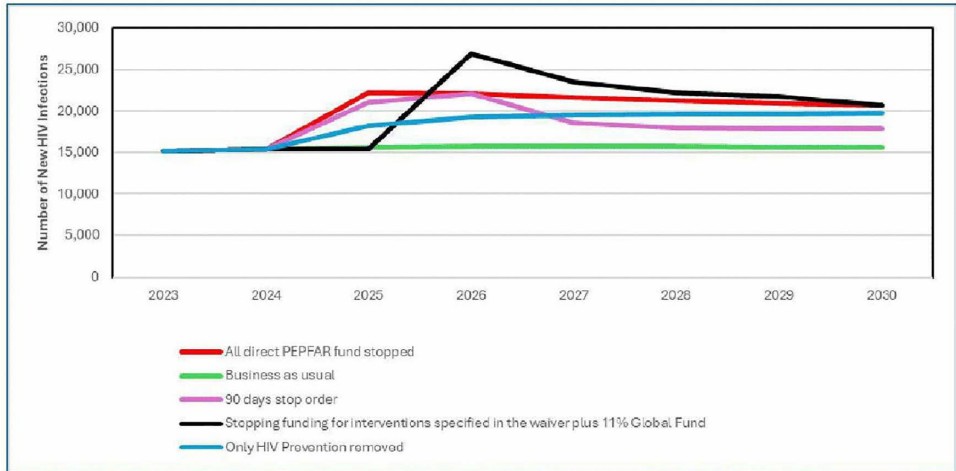

**Fig 1. Trends in new HIV infections by scenario and year.**

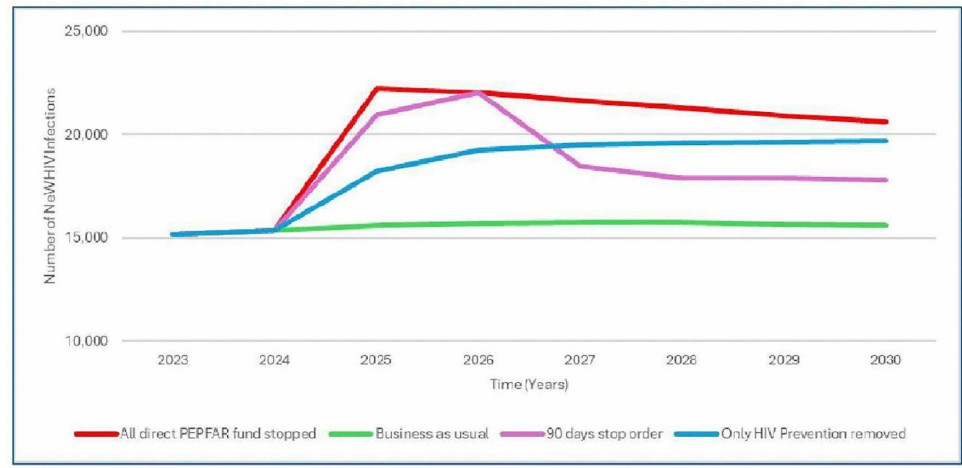

**Fig 2. Trends in new HIV infections by scenario and year (Zoom).**

Stopping funding for interventions specified in the waiver plus 11% reduction in Global Fund funding will increase AIDS-related deaths by more than 13% in 2026 with additional 83,000 deaths through 2030. Stopping all PEPFAR direct funding will result in more than 13% increase in AIDS-related deaths with additional 107,000 deaths through 2030. The 90-day pause and ceasing PEPFAR HIV funded prevention programmes has no significant effect on AIDS-related deaths (Fig 3).

The 90 days pause will reduce adult treatment coverage by 10% from 95% to 85% in 2025, while stopping direct PEPFAR funding will result in reduction in adult ART coverage by 40% to 55% in 2025 and stopping funding specified in the waiver plus 11% reduction in the Global Fund funding will result in 28% reduction on adult ART coverage to 68% in 2026 (Fig 4).

## Discussions

Zimbabwe has a target of reducing new HIV infections by 68% by 2025 from 39,000 in 2018 [5]. The increase in new HIV infection through 2030 by all other scenarios means that the stop work order will defeat the country goal of ending

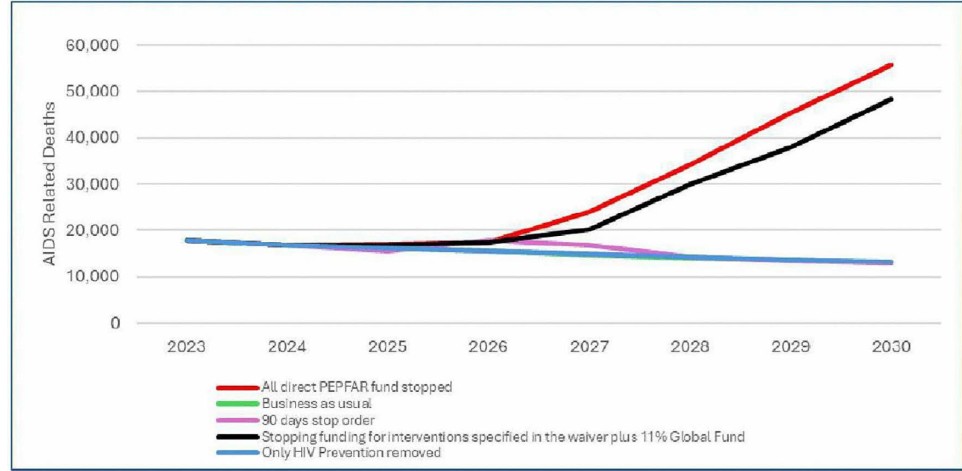

**Fig 3. Trends in AIDS-related deaths by scenario and year.**

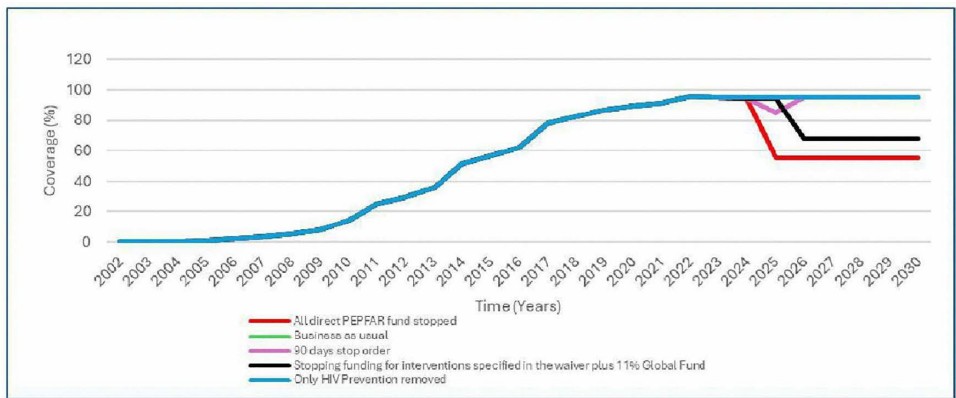

**Fig 4. ART coverage by scenario and year.**

AIDS by 2030. The study found a lower proportional impact of PEPFAR funding interruption on new HIV infections than a recent modelling study performed in South Africa by Ghandi et al which estimated 50% increase in new HIV infections. The increase in new HIV infections is against the global target of reducing new HIV infections by 90% from 2010 baseline [6–8], however UNAIDS estimated that the global targets will require annual resources in 2030 of US$21.9 billion in low- and middle-income countries down from US$29.3 billion, due to efficiency gains in ART, commodities and service delivery modalities [9]

The AIDS-related deaths by scenario estimates increase were lower than the ones from South African modelling study performed by Ghandi et al which estimated 110% increase in mortality [10] because of treatment coverage that is relatively high. The country has a target of reducing AIDS-related deaths by 50% from a baseline of 60,000 in 2018 [5]. The new US administration order will cause the country to fail to achieve the target by increasing AIDS-related deaths as a result of reduced treatment coverage. This concurs with a study done by Jimu C (2025), that states the effects of prolonged funding freeze that will lead to increased AIDS-related death [11]. Globally the targets were set to reduce AIDS-related deaths by 90% by 2030 [12] but the order will increase AIDS-related deaths thereby changing

the epidemic trajectory. The government of Zimbabwe should increase budget allocation for integrated health system response [13] despite the existence of National AIDS Levy which is a domestic financing mechanism that can fill the gap in funding although it will not match the level of funding by PEPFAR. Unmitigated funding reductions can reverse progress in the HIV response by 2030 [14], therefore the government should cover the funding gap. The government also pledged to ring-fence some funds from existing taxes like the sugar, fast foods and alcohol tax. The country is in the process of developing HIV response Sustainability Roadmap that will help the country to mobilise resources to sustain the HIV response gains realized so far since the country may be on a three-to-four-year timeline to completely wind down all PEPFAR support according to the New York times of July 2025. The government of Zimbabwe is negotiating on the conditions of the health Memorandum of Understanding (MoU) with the UG government. The bilateral US – Zimbabwe Health MoU intends to provide life-saving assistance valued at USD345 million. The beneficiaries of the ART programme in Zimbabwe are worried since it is heavily relying on donor funding [15]. In resource constrained environment the government should focus of HIV prevention programmes in order to reduce future treatment cost. The limitations of this study include simplifications, the effect of the funding cuts on targets will not follow a linear approach. We assumed that the proportion change in program coverage will be in line with the PEPFAR funding proportional contribution to the programme.

## Conclusion

This study, using goals model, calibrated to the survey and programme data, found that the new US administration order will result in significant additional new HIV infections and AIDS-related deaths. The additional new HIV infections and AIDS-related deaths will reverse the progress realized so far toward epidemic control assuming there will be no mitigation from government or other donors to close the funding gap.

## Supporting information

**S1 Fig. GOALS conceptual framework.**
(TIF)

**S2 Fig. GOALS model structure.**
(TIF)

## Author contributions

**Conceptualization:** Isaac Taramusi, Amon Mpofu, Tsitsi Apollo, Owen Mugurungi, Benard Madzima, Henry Damisoni.

**Data curation:** Isaac Taramusi, London Makwanya.

**Formal analysis:** Isaac Taramusi, John Stover.

**Investigation:** Isaac Taramusi.

**Methodology:** Isaac Taramusi.

**Project administration:** Isaac Taramusi, Ngwarai Sithole.

**Resources:** Isaac Taramusi.

**Supervision:** John Stover, Simbarashe Rusakaniko.

**Validation:** Isaac Taramusi, John Stover, Ngwarai Sithole, Amon Mpofu, Tsitsi Apollo, Owen Mugurungi, Benard Madzima, Henry Damisoni, Simbarashe Rusakaniko.

**Visualization:** Isaac Taramusi.

**Writing – original draft:** Isaac Taramusi.

**Writing – review & editing:** John Stover, Ali Feizzadeh, Ngwarai Sithole, London Makwanya, Henry Damisoni, Richard Makurumidze.

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
