## [Decision Letter · Decision Letter 0]

23 Nov 2025

PGPH-D-25-01619

The Impact of US Government Stop Work Order on HIV Epidemic Trajectory in Zimbabwe

Dear Dr. Taramusi,

Thank you for submitting your manuscript to PLOS Global Public Health. After careful consideration, we feel that it has merit but does not fully meet PLOS Global Public Health’s publication criteria as it currently stands. Therefore, we invite you to submit a revised version of the manuscript that addresses the points raised during the review process.

We look forward to receiving your revised manuscript.

Kind regards,

Henry Zakumumpa, PhD

Academic Editor

Journal Requirements:

Additional Editor Comments (if provided):

We are pleased to submit our reviewers comments on your manuscript. Please attempt a point by point response to each of the comments raised so that we can move swiftly to a decision. Pay attention to the noted limitations of your study assumptions as observed by the reviewers.

Please be mindful of the language to do with 'Trump orders' and refer to the 'new US administration'.

Reviewers' comments:

Reviewer's Responses to Questions

**Comments to the Author**

1. Does this manuscript meet PLOS Global Public Health’s publication criteria? Is the manuscript technically sound, and do the data support the conclusions? The manuscript must describe methodologically and ethically rigorous research with conclusions that are appropriately drawn based on the data presented.

Reviewer #1: Partly

Reviewer #2: Yes

2. Has the statistical analysis been performed appropriately and rigorously?

Reviewer #1: Yes

Reviewer #2: Yes

3. Have the authors made all data underlying the findings in their manuscript fully available (please refer to the Data Availability Statement at the start of the manuscript PDF file)?

Reviewer #1: No

Reviewer #2: Yes

4. Is the manuscript presented in an intelligible fashion and written in standard English?

Reviewer #1: No

Reviewer #2: Yes

5. Review Comments to the Author

Reviewer #1: The paper makes an important contribution to HIV response sustainability planning in Zimbabwe. The topic is highly relevant and the impact scenarios are useful to consider. The policy recommendations need to be strengthened significantly in the discussion for the findings to be appropriately actionable by government and funding partners. Several other suggestions to improve the paper before publication are contained in the attached report.

Reviewer #2: The research team conducted an epidemiological assessment of the projected impact of the Stop Work Order by the US Government on the projected trends in HIV incidence and projected mortality following the Order. They used a transmission model that uses secondary multi-source and multi-year aggregate data as input to project future incidence of HIV infections. Five modelling scenarios using 2024 baseline outcomes were

used for projecting by scenario-specific HIV burden, i.e., business will continue as usual, 90-day pause of programming followed by resumption, indefinite stopping PEPFAR funded HIV prevention programmes, stopping direct PEPFAR funding including treatment, and worst case scenario whereby all PEPFAR funding and one-third contribution to Global Fund are scrapped. Results suggest that the US Government stop work orders will increase the burden of HIV, i.e., new HIV infections and AIDS related deaths will increase in 2025 and beyond, causing the HIV epidemic to levels last seen in 1998.

Comments

1.In the abstract "waste case scenario" should be stated as "worst case scenario"

2.Is Stop Work Orders referred to as plural or singular? Some places are plural, others singular. Use consistent grammatical number/syntax/morphology.

3.Has the Goals model ever been validated, i.e., how much confidence do we have that the projections resulting from the Goals tool are sufficiently close the true parameters of HIV burden?

4.Does the model generate uncertainty intervals/confidence intervals of the projected estimates of HIV burden? If so, it will be informative for readers to see the range of uncertainty.

5.In Figure 4, the legend does not show the what scenario is represented by the blue line.

6.The discussion appears a bit too lean. Given the magnitude of the looming burden of HIV resulting from the cut in funding for HIV programming, the research team should discuss what the government of Zimbabwe should proactively do. Instead of the government of Zimbabwe waiting (as sitting ducks) for funding to somehow miraculously be reinstated, the research team should offer suggestions. For example, how can the AIDS levy or a significant proportion of the government budget to be redirected to fill the gap in funding? Suggestions for the discussion are: https://www.sciencedirect.com/science/article/abs/pii/S1055329016300401;

7.Another suggestion for the discussion is how to engage with the population so that they also grasp the gravity of the funding cut for HIV programming.

6. PLOS authors have the option to publish the peer review history of their article (what does this mean?). If published, this will include your full peer review and any attached files.

**Do you want your identity to be public for this peer review?** For information about this choice, including consent withdrawal, please see our Privacy Policy.

Reviewer #1: **Yes:** Gemma Oberth

Reviewer #2: **Yes:** Professor Philimon N Gona

 Figure Resubmissions:

---

## [Decision Letter · Decision Letter 1]

17 Feb 2026

PGPH-D-25-01619R1

The Impact of US Government Stop Work Order on HIV Epidemic Trajectory in Zimbabwe

Dear Dr. Taramusi

Thank you for submitting your manuscript to PLOS Global Public Health. After careful consideration, we feel that it has merit but does not fully meet PLOS Global Public Health’s publication criteria as it currently stands. Therefore, we invite you to submit a revised version of the manuscript that addresses the points raised during the review process.

We look forward to receiving your revised manuscript.

Kind regards,

Henry Zakumumpa, PhD

Academic Editor

Journal Requirements:

Additional Editor Comments (if provided):

Reviewers' comments:

Reviewer's Responses to Questions

**Comments to the Author**

1. If the authors have adequately addressed your comments raised in a previous round of review and you feel that this manuscript is now acceptable for publication, you may indicate that here to bypass the “Comments to the Author” section, enter your conflict of interest statement in the “Confidential to Editor” section, and submit your "Accept" recommendation.

Reviewer #1: All comments have been addressed

Reviewer #2: (No Response)

2. Does this manuscript meet PLOS Global Public Health’s publication criteria? Is the manuscript technically sound, and do the data support the conclusions? The manuscript must describe methodologically and ethically rigorous research with conclusions that are appropriately drawn based on the data presented.

Reviewer #1: Yes

Reviewer #2: Yes

3. Has the statistical analysis been performed appropriately and rigorously?

Reviewer #1: Yes

Reviewer #2: Yes

4. Have the authors made all data underlying the findings in their manuscript fully available (please refer to the Data Availability Statement at the start of the manuscript PDF file)?

Reviewer #1: Yes

Reviewer #2: Yes

5. Is the manuscript presented in an intelligible fashion and written in standard English?

Reviewer #1: Yes

Reviewer #2: Yes

6. Review Comments to the Author

Reviewer #1: (No Response)

Reviewer #2: In my initial review submitted on 11/11/2025, I provided comments which the research team did not address or did not receive. I am pasting below the comments for the research team to address:

The research team conducted an epidemiological assessment of the projected impact of the Stop Work Order by the US Government on the projected trends in HIV incidence and projected mortality following the Order. They used a transmission model that uses secondary multi-source and multi-year aggregate data as input to project future incidence of HIV infections. Five modelling scenarios using 2024 baseline outcomes were

used for projecting by scenario-specific HIV burden, i.e., business will continue as usual, 90-day pause of programming followed by resumption, indefinite stopping PEPFAR funded HIV prevention programmes, stopping direct PEPFAR funding including treatment, and worst case scenario whereby all PEPFAR funding and one-third contribution to Global Fund are scrapped. Results suggest that the US Government stop work orders will increase the burden of HIV, i.e., new HIV infections and AIDS related deaths will increase in 2025 and beyond, causing the HIV epidemic to levels last seen in 1998.

Comments

1. In the abstract "waste case scenario" should be stated as "worst case scenario"

2. Is Stop Work Orders referred to as plural or singular? Some places are plural, others singular. Use consistent grammatical number/syntax/morphology.

3. Has the Goals model ever been validated, i.e., how much confidence do we have that the projections resulting from the Goals tool are sufficiently close the true parameters of HIV burden?

4. Does the model generate uncertainty intervals/confidence intervals of the projected estimates of HIV burden? If so, it will be informative for readers to see the range of uncertainty.

5. In Figure 4, the legend does not show the what scenario is represented by the blue line.

6. The discussion appears a bit too lean. Given the magnitude of the looming burden of HIV resulting from the cut in funding for HIV programming, the research team should discuss what the government of Zimbabwe should proactively do. Instead of the government of Zimbabwe waiting (as sitting ducks) for funding to somehow miraculously be reinstated, the research team should offer suggestions. For example, how can the AIDS levy or a significant proportion of the government budget to be redirected to fill the gap in funding? Suggestions for the discussion are: https://www.sciencedirect.com/science/article/abs/pii/S1055329016300401;

7. Another suggestion for the discussion is how to engage with the population so that they also grasp the gravity of the funding cut for HIV programming.

7. PLOS authors have the option to publish the peer review history of their article (what does this mean?). If published, this will include your full peer review and any attached files.

**Do you want your identity to be public for this peer review?** For information about this choice, including consent withdrawal, please see our Privacy Policy.

Reviewer #1: **Yes:** Gemma Oberth

Reviewer #2: **Yes:** Philimon N Gona

Figure Resubmissions:

---

## [Editor Report · Decision Letter 2]

26 Mar 2026

The Impact of US Government Stop Work Order on HIV Epidemic Trajectory in Zimbabwe

PGPH-D-25-01619R2

Dear Isaac Taramusi,

We are pleased to inform you that your manuscript 'The Impact of US Government Stop Work Order on HIV Epidemic Trajectory in Zimbabwe' has been provisionally accepted for publication in PLOS Global Public Health.

Best regards,

Henry Zakumumpa, PhD

Academic Editor